# Who Is the Intermediate Host of RNA Viruses? A Study Focusing on SARS-CoV-2 and Poliovirus

**DOI:** 10.3390/microorganisms12040643

**Published:** 2024-03-23

**Authors:** Carlo Brogna, Domenico Rocco Bisaccia, Vincenzo Costanzo, Gennaro Lettieri, Luigi Montano, Valentina Viduto, Mark Fabrowski, Simone Cristoni, Marina Prisco, Marina Piscopo

**Affiliations:** 1Department of Research, Craniomed Group Facility Srl., 20091 Bresso, Italy; rocco.bisaccia@craniomed.it; 2Institute of Molecular Biology and Pathology (IBPM), National Research Council, 00185 Rome, Italy; vincenzo.costanzo@ibpm.cnr.it; 3Department of Biology, University of Naples Federico II, 80126 Napoli, Italy; gennarole@outlook.com (G.L.); marina.prisco@unina.it (M.P.); 4Andrology Unit and Service of LifeStyle Medicine in Uro-Andrology, Local Health Authority (ASL), 84124 Salerno, Italy; l.montano@aslsalerno.it; 5Long COVID-19 Foundation, Brookfield Court, Leeds LS25 1NB, UK; v.viduto@longcovidcharity.org (V.V.);; 6Department of Emergency Medicine, Royal Sussex County Hospital, University Hospitals Sussex, Eastern Road, Brighton BN2 5BE, UK; 7British Polio Fellowship, Watford WD25 8HR, UK; 8ISB—Ion Source & Biotechnologies Srl., 20091 Bresso, Italy; simone.cristoni@gmail.com

**Keywords:** SARS-CoV-2, poliovirus, gut microbiome, bacteriophage, toxins, vaccine, Sabin

## Abstract

The COVID-19 pandemic has sparked a surge in research on microbiology and virology, shedding light on overlooked aspects such as the infection of bacteria by RNA virions in the animal microbiome. Studies reveal a decrease in beneficial gut bacteria during COVID-19, indicating a significant interaction between SARS-CoV-2 and the human microbiome. However, determining the origins of the virus remains complex, with observed phenomena such as species jumps adding layers to the narrative. Prokaryotic cells play a crucial role in the disease’s pathogenesis and transmission. Analyzing previous studies highlights intricate interactions from clinical manifestations to the use of the nitrogen isotope test. Drawing parallels with the history of the Poliovirus underscores the need to prioritize investigations into prokaryotic cells hosting RNA viruses.

## 1. Introduction

The coronavirus pandemic (SARS-CoV-2, 30 December 2019–5 May 2023) has undoubtedly boosted scientific research in virology, microbiology, and medicine in general. 5 May 2023 was chosen as the date when the WHO declared “the end of the pandemic” [1]. It can be said that many scientists have published a topic or commentary on this virus. From clinical pulmonology, cardiology, hematology, and gastroenterology to neuronal and autoimmune manifestations, all specialties in the medical, biochemical, biological, and biotechnological fields have given their support and contribution. Nevertheless, not all scientists agree on the same topics. Many authors have debated over the origin, intermediate host (bats, pangolins, snakes, or others, such as laboratory artificial creation), and the propagation of SARS-CoV-2.

Considering the perspective of various authors, it is suggested that the pathogen’s transmission may be respiratory, carried through droplets [2,3], while others propose the transmission of SARS-CoV-2 to be orofecal [4,5,6,7]. Notably, a debate arose between Pedersen et al. [8] and Guo et al. [4,5] on this matter. The literature documents a connection between intestinal dysbiosis and coronaviruses affecting animals, with their presence in feces being a well-established feature in the veterinary field, notably confirmed through electron microscopy. Moreover, the orofecal transmission route for these coronaviruses remains firmly established [9,10,11,12,13,14]. In the early 1950s, a similar dilemma arose. After extensive research, Dr. Sabin demonstrated that polio transmission, previously thought to be respiratory, was orofecal. This discovery reclassified Poliovirus as an enteric virus, challenging the long-held beliefs of scientists and the media over the past 50 years. Sabin outlined this pivotal period in one of his papers [15]: “*Does the poliomyelitis virus multiply in or reach the lungs, the trachea, and the nasal mucosa in any significant amounts? It should be pointed out here that the death of the “olfactory pathway” concept of poliomyelitis has not eliminated from consideration the respiratory tract as the chief extraneural system involved in poliomyelitis.*”

It is inferred that, for many years, the respiratory pathway of poliomyelitis and the olfactory epithelium were considered instead of the orofecal route, as he later demonstrated to be the principal route of spread. In addition, Dr. Sabin showed (Table 1) [16] that those vaccinated with the oral vaccine or with natural antibodies did not transmit the pathogen orofecally, unlike those immunized via Salk vaccine injection (see Table 5 of the reference). This anticipates that the critical concept of surface immunity—developed later over the years—where bacteria are also present, is an invaluable resource.

It is worth noting that Poliovirus (PLV) has been isolated from stool samples using antibiotics [17] and that some authors have treated Poliovirus patients with the documented use of antibiotics. [18,19]. Importantly, both transmission routes also spread other pathogens alongside SARS-CoV-2, including commensal bacteria from the respiratory, oral, and intestinal tracts.

In the orofecal transmission of any virus, not only including Poliovirus or SARS-CoV-2, the monitoring of possible sources of spread should be well considered and, among them, the wastewater pathway should not be excluded.

Monitoring the presence and new variants of SARS-CoV-2 through wastewater surveillance is, therefore, crucial [20].

On the other hand, wastewater is also being studied because of the presence of many dangerous bacterial species or the growing emergence of bacterial resistance [21] and almost every state conducts wastewater monitoring.

The indications of a bacterial species’ involvement in the pathogenesis of SARS-CoV-2 infection have been numerous during the past four years and the scientific evidence lays an important foundation for it.

Many studies have shown an essential connection between dysbiosis of the intestinal microbiome following SARS-CoV-2 infection. In particular, the persistence of microbiome alterations in both long COVID-19 [22,23] and major opportunistic infections during the acute phase were observed [24]. More directly, the presence of certain bacteria and their metabolism is a non-negligible aspect in COVID-19 patients.

In addition to observations on dysbiosis or associated outbreaks of harmful bacteria, a diverse range of symptoms have been reported in sick individuals—that have been analyzed both metabolically and neuronally—which underscore the suspicion of a bacterial cofactor in SARS-CoV-2 infections.

The literature contains symptomatic and even asymptomatic cases manifesting genetic detection on the nasopharyngeal testing of SARS-CoV-2. In the symptomatic cases, we might delineate that some sort of equilibrium occurs in the most crucial area—the overlying epithelial–microenvironment interface.

Suspicions of bacterial class involvement are also present in various papers which report important data during the genetic analysis of samples derived from nasopharyngeal swabs.

In fact, some authors have clarified that nasopharyngeal samples contain subgenomic sequences of SARS-CoV-2 [25]. They note the following: *‘The results described here fully support that SARS-CoV-2 genomic and subgenomic RNAs are present in diagnostic samples even in late infection/after active infection’ and ‘The detection of subgenomic RNA is therefore not direct evidence of active infection; instead, its presence at lower levels than virion genomic RNA results in detection for a shorter period unless using, e.g., highly sensitive NGS?”.* It should be pointed out that the authors have described nasopharyngeal sampling as a viral RNA collection event and readers and scientists should, therefore, reflect that the sampling is performed on mucous membranes that are overabundant with bacteria and the fact that subgenomic sequences in non-SARS-CoV-2 sufferers could also have bacteria as a reservoir cannot be excluded [25]. In light of the observations thus far, it seems clear that the microenvironment overlying all mucous membranes plays a key role in the pathogenesis of infections of defined RNA viruses such as Poliovirus and, in our case, SARS-CoV-2. The mucous membranes of interest include nasal, oropharyngeal, ocular, alveolar, and intestinal. In the case of SARS-CoV-2, it is not uncommon to observe ocular manifestations such as conjunctivitis and lacrimation, in addition to colds and sore throats [26,27,28]. Some types of pneumonia and relatedsymptoms, from gastrointestinal to neurological and even ocular, had already been observed with other coronaviruses (HCoV-NL63) [29].

It should be pointed out that despite the controversies relating to the epidemiology and the manifestation of very heterogeneous clinical aspects, everyone agrees SARS-CoV-2, like other RNA viruses (Poliovirus, HIV, and H1N1), is typically found on mucous membranes. Moreover, their first moment of adhesion occurs on epithelial cells, with some concerns emerging about the actual mechanisms of infectivity described by many scientists; one can consider that in front of the epithelial cells there is a thick layer of bacteria.

Current in vitro models show that SARS-CoV-2 binds to angiotensin-converting enzyme 2 (ACE2) receptors and the transmembrane serine proteinase 2 (TMPRSS2) receptor found on epithelial cells. From there, it initiates the process of endocytosis, followed by the subsequent invasion, replication, and spread of the virus. These pathways can also be described for other viruses, with other receptors; however, it excludes an important actor—bacteria in the microbiome.

The numerical ratio of epithelial cells to bacteria in the microbiome is clearly in favor of the latter and this fact should be taken into account. In simpler terms, is it certain that RNA viruses can easily pass through the microbiome layer without any hindrance? Have researchers confirmed that bacteria do not alter their metabolism when exposed to a new viral pathogen? Are bacteria entirely excluded from the processes of infection, replication, and mutation of the pathogen? We proposed a possible molecular mechanism of bacteria infection by Sars-CoV-2 illustrated in Figure 1.

The observation above prompted an investigation beyond conventional testing methods, whereby controlling bacteria and their interaction with SARS-CoV-2 revealed adverse effects.

In this paper, we outline the results of previous studies and discuss their significance in light of the current knowledge. We summarize the methods used that enabled the reclassification of the bacteriophage SARS-CoV-2 pathogen. Through an examination of the techniques (immunofluorescence microscopy, molecular genetics, electron microscopy, proteomics, and the use of radioisotopes), we illustrate the useful pathway to include or exclude RNA viruses as bacteriophages. In this paper, we also report the latest data on the bacterial replication and lysogenic aspects of Poliovirus, emphasizing the importance of oral vaccination, as adopted by Dr. Sabin.

## 2. Overview of the Results on the Bacteriophage Behavior of SARS-CoV-2

### 2.1. Initial Molecular Investigation of Viral RNA in Bacterial Cultures

Previously, Petrillo et al. [30] demonstrated, through the Luminex genetic test, an increase in SARS-CoV-2 viral RNA in bacterial cultures derived from fecal samples of COVID-19 patients, along with contamination of similar samples from non-infected individuals. *As reported by the authors in [30], the nucleic acid extractions were performed using the NucliSe s^®^ easyMAG™ extraction system (bioMerieux, Marcyl’Étoile, France) and the extracts were stored at −80 °C until use. The determination of the viral RNA load was performed utilizing Luminex technology (Life Technology, USA, see Dunbar* [31] *for an overview). The detection was performed using the NxTAG^®^ CoV Extended Panel, a real-time reverse transcriptase PCR assay detecting three SARS-CoV-2 genes on the MAGPIX^®^ NxTAG-enabled System, and the AccuPlex™ SARS-CoV-2 Reference Material Kit (SeraCare) as reference standard with sequences from the SARS-CoV-2 genome. Multiplex plates were produced in-house and RNA tags were linked with the NaxPhot reagent before the analysis. Multiplex plates were transferred to the 37 °C pre-heated MAGPIX heater and the signal was acquired using the xPONENT and SYNCT software (Luminex Molecular Diagnostics-http://www.luminexcorp.com/). Each running batch handled up to 94 clinical specimens plus the positive and negative experimental controls. The total turnaround time was around 4 h. Luminex detection was reported in arbitrary units, following Floridia* et al. [32]: *ARB = [Σ(SI × NF × COF)/Σ (NF × COF)], where SI: signal intensity converted to counts/s; NF: noise factor; COF: correction Ffactor.*

Arbitrary units are derived from a normalization coefficient, eliminating dimensions and stabilizing signal oscillation. This method is crucial in analytical fields to achieve clearer and more comparable signals [33]. Absolute units are often highly unstable and are rarely used for quantitation. Luminex technology also utilizes arbitrary units, particularly in multi-residual analysis [34].

The authors observed the emergence of viral mutations during bacterial cultures, as documented in their study. It is intriguing to investigate the reasons behind bacteria generating new virus variants, potentially linked to the fundamental role bacteria play in supporting life on our planet. Additionally, in other studies [35,36,37,38], researchers observed the metabolic activity of bacteria, identifying certain products, termed toxin-like peptides (Ps), present in the plasma of infected individuals. These peptides, produced from bacterial cultures, exhibited a potential half-life of up to 30 days. Particularly fascinating is the creation of sample C, as described in these studies [35,36,37,38], where the rate of these peptides remained unchanged compared to other samples. However, the decrease in viral load shows that it almost represents a bacterial defense mechanism in the presence of a new pathogen. Three other papers [39,40,41] show what happens when bacteria from the gut microbiome interact with the SARS-CoV-2 virus and with the Poliovirus (PLV).

Data was obtained using microscopy provide a cross-reference with those data presented in the first paper, in which an increase in viral RNA was observed in bacterial cultures 30 days after initiation, using Luminex [30].

Finally, the authors agree on a much faster method of determining bacterial involvement during virus propagation—a versatile test using the nitrogen isotope (^15^N) in bacterial cultures contaminated with SARS-CoV-2 and PLV [39,40,41]. The same procedure for stool collected from SARS-CoV-2-infected patients, and its proteomic analysis was performed for bacterial cultures sourced from the stool of subjects with terminal poliomyelitis [41].

### 2.2. Use of Electron Microscopy and Immunofluorescence Microscopy as Confirmation of the Bacteriophage Aspect of SARS-CoV-2

It is intriguing to examine the electron microscope images and immunofluorescence findings shared by the authors. The data strongly suggest that contrary to prevailing beliefs, bacteria serve as the primary intermediate host.

We have reported, just as has been published, typical cases of SARS-CoV-2 derived from bacteria found in stool samples [39,40].

Figure 2, Figure 3 and Figure 4 are described by the authors in [39,40]. They show examples of many investigations obtained in the bacterial culture derived from SARS-CoV-2-infected patients (Panels A–G). Panel H shows the presence of the nitrogen isotope, ^15^N, in the spike protein of SARS-CoV-2 after seven days of bacterial culture, where the isotope was added to the nutrient broth.

Figure 3, as described by the authors, presents the immuno-EM pre-embedding technique as a control to the immunogold post-embedding assay and the immunogenicity testing of the nucleocapsid SARS-CoV-2 (N) protein (Abcam, #ab273167). Antibody testing was performed with the GFP-tagged N protein on Hela cells.

Immunofluorescence microscopy (Figure 4, panels A–D—Zeiss Axioplan 2, Axiocam 305 color, magnification 100×), as described by the same authors, was performed according to the manufacturers’ protocol, using primary antibodies against the SARS-CoV-2 nucleocapsid protein “Sars Nucleocapsid Protein Antibody [Rabbit Polyclonal]—500 μg 200-401-A50 Rockland” and the “Goat anti-Rabbit IgG (H+L) Cross-Adsorbed Secondary Antibody, Cyanine3 #A1052” as a secondary antibody. Gram-positive bacteria were stained with a primary antibody (Ab (BDI380), GTX42630 Gene Te”) and “Goat anti-Mouse IgG (H+L), Super-clonal™ Recombinant Secondary Antibody, Alexa Fluor 48” as a secondary antibody. The images confirm the presence of SARS-CoV-2 particles (red light in the fluorescence images) in relationship with the bacteria (green light in the fluorescence images). The control of the specific reactivity of primary antibodies versus Gram-positive bacteria was performed using a culture with a negative molecular test for SARS-CoV-2. It is important to search in the literature if other authors have tested the same antibody and validated it. The control of the specific reactivity of antibodies versus Gram-positive bacteria was assumed from the worl of Kohda et al. and Kameli et al. [42,43]. In addition, the Gram-positive bacteria antibody [BDI380] Genetex is a mouse monoclonal antibody that has been validated and tested to be reactive toward many Gram-positive bacteria—precisely what we were interested in finding out. The specifications can be found at the following link: https://www.genetex.com/Product/Detail/Gram-Positive-Bacteria-antibody-BDI380/GTX42630 (accessed on 20 February 2024). The authors stated the following: “*Reactive with lipoteichoic acid (LTA) of many Gram-positive bacteria. Cross-reacts with Listeria monocytogenes (all serotypes), Streptococcus pneumoniae, Staphylococcus aureus, Staphylococcus epidermidis, Enterococcus faecium, Bacillus cereus, Bacillus subtilis and group B Streptococcus (weak). Does not react with Clostridium perfringens”*. The control of the specific reactivity of primary antibodies versus the nucleocapsid protein of SARS-CoV-2 was assumed by Zhao et al. [44]. In addition, the Rockland primary antibody (Sars Nucleocapsid Protein Antibody [Rabbit Polyclonal]—500 μg 200-401-A50 Rockland) we used for immunofluorescence experiments has no less than 76 references where it has been used, and its remarkable specificity towards SARS-CoV-2 was defined. The references can be seen on the following page: https://www.rockland.com/catego-ries/primary-antibodies/sars-nucleocapsid-protein-antibody-200-401-A50/#productReferenceSectionWrapper (accessed on 20 February 2024).

### 2.3. Analysis and Exclusion of Other Bacteriophages Present in Bacterial Cultures

Upon further investigation, the authors confirmed the coronavirus-like nature of the shapes observed in the electron microscopy images by excluding other forms of bacteriophages, which are described with distinct shapes and tails, unlike the viral images presented. Table 2 provides a list, obtained through proteomic profiling via mass spectrometry [32,45,46,47], of other bacteriophages present at the end of the 30-day cultures. These bacteriophages differ in size, shape, and notably, all possess a tail, which is absent in the viral particles depicted in the TEM and immunogold images. Specifically, each bacteriophage identified in the data exhibits a structured tail, either long or short, and an icosahedral form, contrasting with the coronavirus-like shapes observed in the TEM and immunogold images.

### 2.4. Evidence of Bacterial Lysis on a Petri Dish

In their initial paper [38], the authors utilized NGS sequencing data to observe a list of bacteria present in certain amounts before initiating the culture. After contaminating the same cultures with viral particles and waiting for 30 days, they noticed a decrease in some beneficial bacteria in the microbiome, while others increased. This discovery was corroborated by other reports in the literature, which also demonstrated a decrease in the same class of bacteria. In light of this, the authors decided to select some bacteria (Dorea, in this case), perform growth tests on a rigid medium, and check for lytic plaques after contamination with a supernatant containing SARS-CoV-2. The NGS sequencing data experiments were performed as follows: *At day 0 (B0), DNA from fecal samples was extracted with the E.Z.N.A.^®^ Stool DNA Kit (Omega Bio-Tech, Norcross, GA, USA). Two types of kit for the nucleic acid extractions from samples after 30 days (B1) of culture were used, the MasterPure Complete DNA and RNA purification kit (Lucigen, WI, USA) and the PureLink Viral RNA/DNA kit (Thermo Fisher Scientific, Waltham, MA, USA). All extractions were performed following the manufacturer’s recommendations. The Ovation^®^ Ultralow V2 DNA-Seq Library Preparation kit (NUGEN, San Carlos, CA, USA) was used for library preparation, following the manufacturer’s instructions. Both the input and final libraries were quantified with the Qubit 2.0 fluorometer (Invitrogen, Carlsbad, CA, USA) and the quality was tested using the Agilent 2100 Bioanalyzer High Sensitivity DNA assay (Agilent Technologies, Santa Clara, CA, USA). Libraries were then prepared for sequencing and sequenced on a NovaSeq 6000 in 150 bp paired-end mode. Bioinformatic reconstruction was calculated on UniProt’s fasta file of bacterial and bacteriophage sequences.*

In Figure 5, the authors depicted lytic plaques observed on a plate inoculated with a bacterium of the genus Dorea, contaminated at various sites with varying amounts of supernatant derived from a 30-day bacterial culture. Molecular testing confirmed the presence of SARS-CoV-2, as detailed in [40]. The authors reported the number of sequences in GISAID within the fragment extracted from the plaque, enabling the identification of SARS-CoV-2 in the lytic lesions.

### 2.5. The Use of Radioisotopes to Ultimately Confirm the Reclassification of RNA Viruses as Bacteriophages

In the history of medicine and biology, radioisotopes or radionucleotides have often been used. It is possible to recall the famous Hershey–Chase experiment, which demonstrated genetic material is DNA and not protein, using radioactive phosphorus (^32^P) and sulfur (^35^S). Nitrogen in the molecular form, N_2_, is the most abundant element in the Earth’s atmosphere, constituting about 78% of it. There are two stable isotopes, ^14^N (99.63%) and ^15^N (0.37%). They have a stable half-life, while the other isotopic forms are unstable and have short half-lives. The ^15^N isotope has a relative atomic mass of 15.0001u compared to the ^14^N isotope, which has a relative atomic mass of 14.0067u, so it is heavier as well as having a different atomic spin. Nitrogen is present in major biochemical molecules such as nucleic acids, proteins, and some vitamins. It was used in classical molecular biology experiments to show the double helix of DNA. As was carried out by the authors of [38,40], for SARS-CoV-2, by adding the nitrogen ^15^N isotope (Figure 2H) to the bacteria culture broth, its bacterial replication was demonstrated. In the last study [41], the authors themselves show how the same procedure for the feces of COVID-19 patients (the culture of bacteria with added ^15^N) was analyzed for bacterial cultures, derived from poliomyelitis patients who were terminally ill and contracted the disease at a young age.

As also expected for Poliovirus, its proteins are expressed with the nitrogen isotope, showing that the pathogen remains present in the microbiome of poliomyelitis patients for a long time. In culture conditions, it is replicated by the bacteria of the gut microbiome (Figure 6). These data will still need to be analyzed in light of metabolic and toxicological aspects, as they have already been discussed in relation to SARS-CoV-2.

## 3. Discussion

### 3.1. The Evaluation of the Integration of Microscopic, Genetic, and Proteomic Data on SARS-CoV-2 to Determine the Major Cells Infected

The intermediate host of viruses from the coronaviridae family is not exclusively human. Extensively studied and documented in the animal kingdom, these pathogens prompt critical fecal analysis from a veterinary perspective [48,49,50,51]. Symptoms in both humans [52] and animals often manifest in the gastrointestinal tract, characterized by mucosal alterations, leading to frequent diarrhea [48,49,50,51]. Evidence ranging from decreased beneficial bacteria in the microbiome to virus presence in sewage, even during warmer months, points to bacteria as an intermediate host. Symptoms among patients may not solely stem from viral infection but may involve bacterial involvement and metabolism. While symptoms like conjunctivitis and gastrointestinal issues, including diarrhea, are not directly linked to viral infection, they have been reported as manifestations of COVID-19 in some studies [26,28,53]. These symptoms have also been observed with other coronaviruses (HCoV-NL63) [29]. The “Arcturus” variant of SARS-CoV-2 is also particularly impactful with these two symptoms in children (https://www.cnbctv18.com/healthcare/COVID-new-symptom-variant-arcturus-omicron-conjunctivitis-itchy-eyes-children-16399051.htm) (accessed on 20 February 2024) [53].

Examining the images depicting the intestinal mucosa of certain animals and their association with coronavirus particles yields intriguing insights. For instance, in a study [54], researchers demonstrated the close interaction with the intestinal microenvironment in cattle affected by bovine coronavirus (B-CoV). Additionally, autopsy preparations of sacrificed animals’ intestinal mucosa revealed coronaviruses within the intestinal lumen and their interaction with surrounding structures (see Figure 2 of [54]). (https://www.ncbi.nlm.nih.gov/pmc/articles/PMC7130198/pdf/main.pdf, accessed on 20 February 2024).

Examination of microscope images reveals the presence of viral particles within bacteria from the intestinal microenvironment. It is worth noting that bacteria can be easily distinguished from multivesicular bodies (MVBs) or vesicles with intracytoplasmic double membranes. Given the abundance of bacteria of varying shapes [55] and sizes, and considering that the microscopic section may capture an oblique, apical, or coronal portion of bacterial rods within the pellet, MVBs are exclusive to the intracytoplasmic formations [56] of human cells, as are vesicles with double membranes [57]. The authors [38,39,40] performed cultures exclusive to bacteria and after 30 days no eukaryotic cells were present or could survive. Electron microscopy images of bacteria can be found in these studies [58,59,60]. Baker et al. in 1982 [61] and Mathan et al. in 1975 [62] emphasized the presence of various forms of the same coronavirus, defining it as pleiomorphic. Furthermore, Orenstein et al. [63] noted that coronavirus particles can be observed at different densities, particularly ribonucleoprotein. The authors state the following: *The mature particles had a central spherical core, which was either clear, contained electron-dense granules (nucleocapsid), or were totally electron-dense (Figure 2D, 33—*https://www.ncbi.nlm.nih.gov/pmc/articles/PMC2763395/pdf/nihms100890.pdf, accessed on 20 February 2024) [63].

The viral particles that highlight coronavirus seem to be identical to those shown in Figure 2A,B in the present manuscript. Furthermore, in Figure 2A, it is possible to see the viral particles (1–4) within the bacterium, with different densities from left to right. Bullock et al. [64] show, precisely, in them Figures 1c and 2c (https://www.ncbi.nlm.nih.gov/pmc/articles/PMC7825881/pdf/main.pdf, accessed on 20 February 2024), viral particles being clearly visible—indicated by the authors as coronavirus particles and we similarly associate those we show inside the bacterium in Figure 2A and around the lysed bacterium in Figure 2B. The authors quote the following sentence demonstrating that the spike protein, unless specially stained (e.g., with tannic acid), is rarely visible in microscopic images as they report: “(ii) *Coronaviruses do have projections on the surface; however, in thin sections, the “spikes” on the outside are not always (indeed, not usually) clearly visible, unless specially stained (e.g., with tannic acid). They may or may not appear as a very short ‘fuzz*’”.

To reach these conclusions, a single test is not always sufficient and the comparison of images with what has already been produced in the literature is very important.

Indeed, sometimes electron microscopy alone is not enough to be sure of the data displayed. Therefore, the combination of a series of tests, where one can act as a control for the other, becomes important. These range from antibody validation to complementary tests such as genetic, proteomic, and radioisotope tests.

A validity check of antibodies is important, as shown in Figure 3A–D (primary antibody against Nucleocapsid SARS-CoV-2 (N) protein (Abcam, #ab273167)). Figure 3E,F show that the validated antibodies bind to target proteins within the bacteria.

Furthermore, for the embedding technique (Figure 2C) of electron microscopy, the rabbit monoclonal antibody [EPR24334-118] to the SARS-CoV-2 nucleocapsid protein antibody (Abcam ab 271180) was used. This antibody was developed by the collaboration of Dr. Hangping Yao, State Key Laboratory for the Diagnosis and Treatment of Infectious Diseases in Zhejiang University, School of Medicine. It has received three excellent reviews from verified professional scientists https://www.abcam.com/products/primary-antibodies/rabbit-monoclonal-epr24334-118-to-sars-cov-2-nucleocapsid-protein-ab271180.html?productWallTab=Abreviews, accessed on 20 February 2024) and it has already been approved in [65,66].

Nevertheless, the virions presented in Figure 2A,B in this manuscript are identical to the coronaviruses (SARS) present in Figure 2D of study [63] (https://www.ncbi.nlm.nih.gov/pmc/articles/PMC2763395/pdf/nihms100890.pdf, accessed on 20 February 2024). *Electron dense granular nucleocapsid material can be seen in some of the virions (arrows) or* in Figure 1A–E of study [67] (https://www.ncbi.nlm.nih.gov/pmc/articles/PMC3322934/pdf/03-0913.pdf): *Nucleocapsids measure 6 nm in diameter and are mostly seen in cross-section. Some virions have an electron-lucent center, with the nucleocapsid juxtaposed to the envelope, while others are relatively dark when the nucleocapsid is present throughout the particle.*

Immunofluorescence images (Figure 4A–E) of the sample highlight Gram-positive bacteria and the nucleocapsid protein of SARS-CoV-2 in the merge phase. The choice of a healthy subject for the negative control is critical as already argued in [38], because, unfortunately, SARS-CoV-2, with bacteriophage behavior, possesses both lytic and lysogenic characteristics and it is expected that many people infected and then cured may continue to show traces of the virus even after some time.

Immunofluorescence has been helpful in beginning to understand the phenomenon of bacteriophage behavior, and the use of already tested products is critical [65,66,67,68].

It is crucial to emphasize that when considering the use of viral-tagged antibodies, in microscopy within a bacterial culture and their specificity, it is important to note that these antibodies produced against viral pathogens, which have entry points on the mucosal surface, tend to localize in the epithelium as well. In this area, the antibodies, whether generated by vaccines or acquired through natural immunity, act to prevent binding between the virions and the cell wall (as observed in the case of SARS-CoV-2 between the ACE 2 and spike proteins). This concept necessitates that the virus antibodies, in general, must operate on the epithelial surface, where millions of bacteria are present. Also, the choice of antibodies against bacteria is essential. The Gram-positive bacteria antibody [BDI380] Genetex is a mouse monoclonal antibody that has been validated and is tested to be reactive to many Gram-positive bacteria—exactly what the authors in [38,39,40] were interested in finding out [69].

The quantification or presence of the genetic aspect is required in accordance with current opinion. The data presented in Figure 2 and Figure 4 comes from a 30-day culture under anaerobic/semi-aerobic conditions of a bacteria sample from the fecal collection of a subject with SARS-CoV-2-positive stool. Through Luminex data, expressed in au (arbitrary units), the authors in [30] observed an increase in RNA growth. The au in Luminex results from a normalization coefficient that eliminates the dimension and stabilizes signal oscillation. This approach is considered fundamental in the analytical field for obtaining cleaner and more comparable signals [33]. Absolute units can be highly unstable and their use for quantitation is rare. Arbitrary units are also employed with Luminex technology, especially in multi-residual analysis [34].

Parallel to the genetic aspect proposed by Petrillo et al. [30], the authors in [35,36,37,38,39,40] demonstrated, using fluorescence microscopy analyzing slides of bacterial culture samples at predetermined intervals up to 30 days, that the bacteriophagic characteristic of SARS-CoV-2 can be highlighted. The manuscript in [39] demonstrates how the protein increase follows the genetic increase in the bacterial culture with SARS-CoV-2.

In a sample from a 30-day bacterial culture, the presence of other bacteriophages was assessed through mass spectrometry examination to confirm protein expression. These data were compared with the literature findings, revealing the presence of other bacteriophages with distinct shapes and featuring a tail. This serves as an indirect control measure, aiding in the accurate interpretation of electron microscopy images. Previous studies have also noted the mutual exclusion of bacteriophages attempting to replicate in bacteria infected with different types of viruses [70]. “*The term mutual exclusion is used to designate an extreme form of interference which often occurs when two phage particles attack the same host cell. The cell liberates numerous particles of one of the parental types and not a single particle of the other parental type*. *Exclusion of phage from a multiplication in a bacterium infected a few minutes earlier with a similar active or ultraviolet-killed phage has been noted repeatedly (Luria and Delbruck, 1942; Luria, 1945; Delbriick and Bailey, 1946; Luria and Dulbecco, 1949”).*

What is continually ruled out by many researchers is the possible production of different and usual metabolites or proteins by microbiome bacteria. In Figure 7, we schematize what our early evidence reported. Bacteria produce molecules with toxic-looking amino acid structures.

The lytic plaque (Figure 5), while not intended for quantitative analysis but solely for investigative purposes, can guide the collection of samples to verify the presence or absence of SARS-CoV-2 mutations in the bacterial culture. This verification can be conducted using NGS sequencing systems, as detailed in [38].

### 3.2. Considerations on the Importance of Radioisotope Testing as the Most Immediate Investigation

Beyond mere speculation, the question arises as to whether a virus classified as human can exhibit bacteriophage behavior, which can be investigated through a simple experiment using radioisotopes. This experiment enables any laboratory to determine whether pathogen proteins display the heavier nitrogen-15 (^15^N) isotope after a specific period of bacterial culture, thereby establishing that viral proteins result from reproduction within the bacterial cell. The Meselson–Stahl experiment [71] provided researchers with insights into DNA replication, offering a fundamental understanding of genetic phenomena related to heredity and disease, all based on the utilization of nitrogen isotopes. This methodology represents a significant milestone in biology and biochemistry; however, it is important to note that the use of ^15^N nitrogen cannot independently corroborate microscopy data.

## 4. Conclusions

In light of what has been observed so far, the question remains as to whether a historical RNA virus has the same bacteriophage behavior. Among the various viruses, the one that is best known and resolved by Sabin vaccination instead of injection vaccination is the Poliovirus. The different vaccination route - oral with attenuated virus - suggested a similar behavior. Figure 6 shows the observation (the first in the world, to the best of our knowledge) that Poliovirus also infects bacteria in the human microbiome. We expected to find that in a severe poliomyelitis patient, it was chronically present in the subject’s stool. The analysis first checks prokaryotic cells—their involvement suggesting a different narrative to what has been described in the literature. A viral pathogen having bacterial involvement, as demonstrated for SARS-CoV-2 and similarly with the nitrogen isotope experiment for Poliovirus, described here, allows the following aspects to be emphasized: (1) the pathogen’s intermediate host is not animals, but bacteria; (2) the main route of spread is orofecal; (3) the involvement of wastewater, as observed in many studies, suggests the bacterial vector; (4) geographic areas with higher rainfall density are most affected; (5) finding an inert strain that can serve as a base for mass vaccination may be the best and immediate solution as observed by Sabin; (6) clinical conditions of the ill may be an expression of the toxicological rate of the microbiome as observed in [35,36,37,38,39,40]; and (7) vaccination or sensitization of bacteria to the viral pathogen is the gold standard (concept of mutual exclusion).

Sabin left clues that were probably underestimated. In one of his papers [16], he shows the following table (Table 1), which we reproduce in italics. From Table 1, we see that only those who had allowed the human microbiome to encounter the pathogen prevented its subsequent replication and spread, showing that the surface immunity that occurs between epithelial cells and bacteria is vastly superior.

In light of these observations, we believe that more studies should be conducted from the perspective of the microbiome and viral pathogens. Many more investigations should be performed on the toxicological pictures emerging from the microbiome, as we have already observed in some works and RNA viruses.

It is time to correct the mistakes of the past and control the bacteria in viral infection mechanisms first.

## Figures and Tables

**Figure 1 microorganisms-12-00643-f001:**
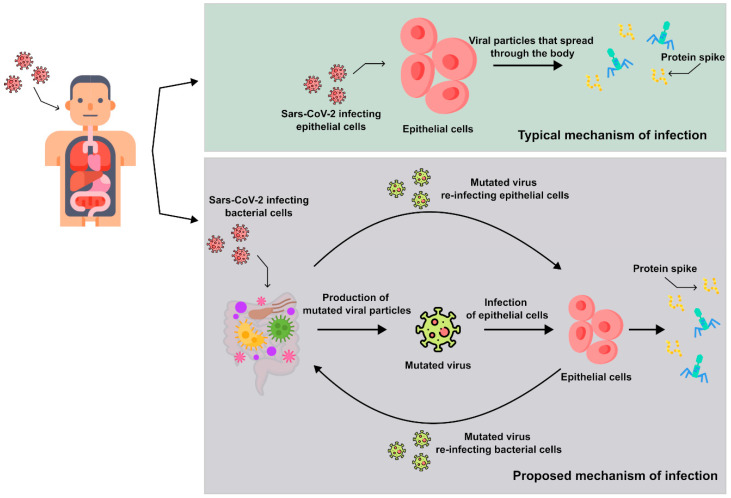
In the upper box, the conventional mechanism widely recognized thus far is illustrated, depicting the direct attachment of the virus to epithelial cells. In the lower inset, other studies unveil an alternative mechanism, where the virus primarily adheres to bacteria within the human microbiome (oral, alveolar, or intestinal), where it subsequently undergoes replication and mutation. The virion then sequentially or concurrently attaches to and invades surface cells. This mechanism also justifies why RNA virions continually mutate to the extent that antivirals do not always work in the treatment of the disease in the long term (a typical example of this information is therapy that is continually modified with antivirals in HIV carriers).

**Figure 2 microorganisms-12-00643-f002:**
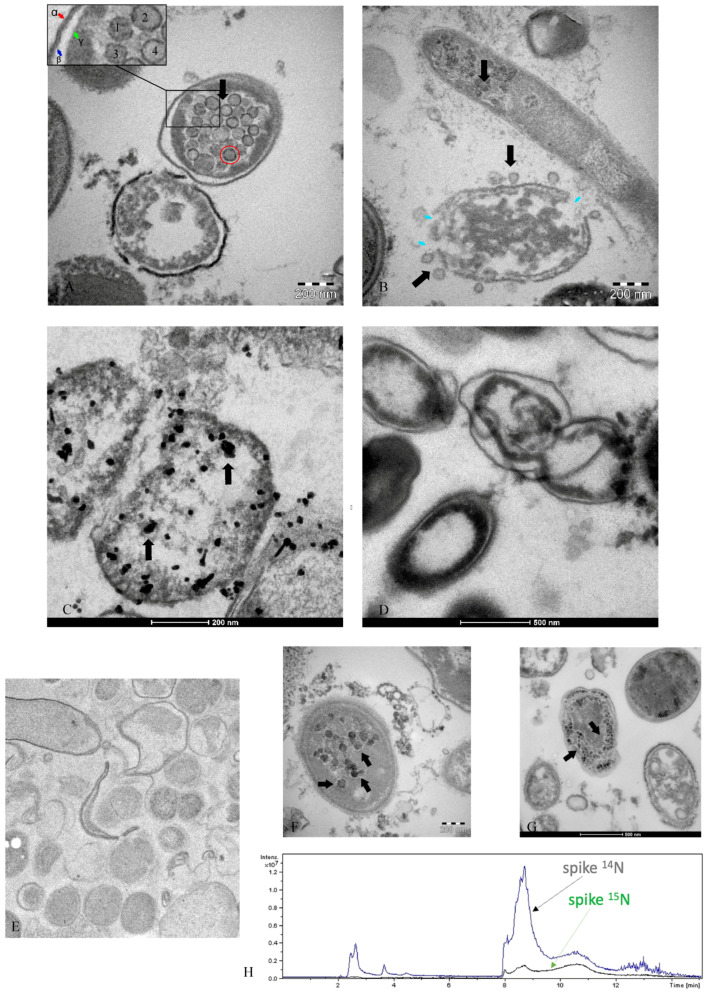
Micrographs of SARS-CoV-2 viral particles within bacteria from bacterial culture, as reported by the authors in [34,35,36] who we thank. Transmission electron microscope images ((**A**,**B**)—Tecnai G2 Spirit BioTwin, FEI, equipped with a VELETA CCD digital camera (Olympus Soft Imaging Systems GmbH, Münster, Germany)—of SARS-CoV-2 (black arrows) inside a bacterium (**A**) and outside a matrix resembling the extracellular lysate of a bacterium (**B**), the blue arrows indicate the breaking points of the bacterial wall. (**A**) numbers 1–4 present typical viral particles, α and β; the cellular wall of bacterium; and γ inner membrane. Post-embedding immunogold. Inside the red circle is a typical coronavirus particle. (**C**) Post-embedding immunogold—rabbit monoclonal to SARS-CoV-2 nucleocapsid protein antibodies ligating to the secondary anti-rabbit antibody (10nm, gold-conjugated), indicating the virus inside the bacteria of the gut microbiome (Tecnai G2 Spirit BioTwin, FEI, equipped with a VELETA CCD digital camera (Olympus Soft Imaging Systems GmbH)). (**D**) shows the negative control of bacterial stool culture of a healthy person after 30 days, without primary antibodies and only with the secondary antibody. Micrograph (**E**) shows the negative control (scale bar 500 nm) with primary and secondary antibodies in cultures derived from healthy patients. (**F**,**G**) show bacteria with viral particles inside them. (**H**) shows the peptide mapping of the SARS-CoV-2 spike protein. Peptide mapping of SARS-CoV-2 spike protein was acquired through liquid chromatography–mass spectrometry associated with ^14^N and ^15^N profiles and was performed on an aliquot of a SARS-CoV-2-positive sample.

**Figure 3 microorganisms-12-00643-f003:**
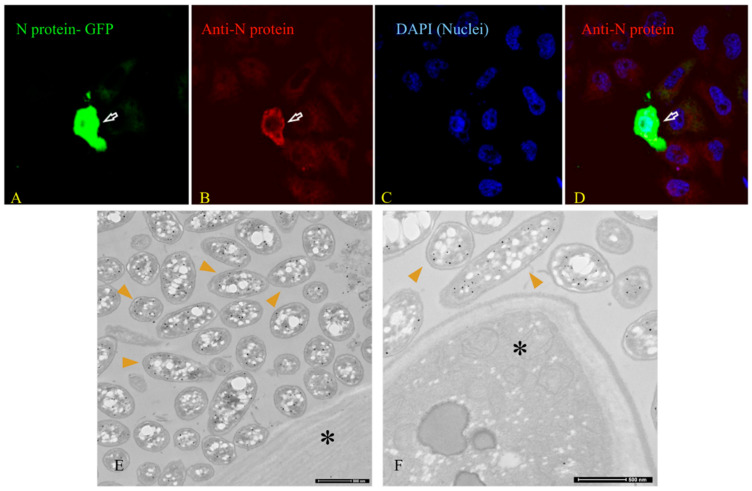
(**A**–**D**) Immunofluorescence images ( magnification 100×) show the validation of the antibody against the nucleocapsid protein of SARS-CoV-2 HeLa cells, which were transfected with the GFP-tagged N protein of SARS-CoV-2, fixed, and then labeled with an antibody against the N protein of SARS-CoV-2. The white arrows indicate that cells expressing GFP-tagged N protein of SARS-CoV-2 were labeled with antibodies against the N protein. (**E**,**F**—scale bar 500 nm), **pre-embedding immunogold**, show bacterial cultures with the SARS-CoV-2 protein N-binding antibody both around and inside the bacteria. The asterisk (*) indicates the only one eukaryotic yeast we found in the cultures, with no viral particles inside. With thanks to the authors of [39,40] for permission to use their images.

**Figure 4 microorganisms-12-00643-f004:**
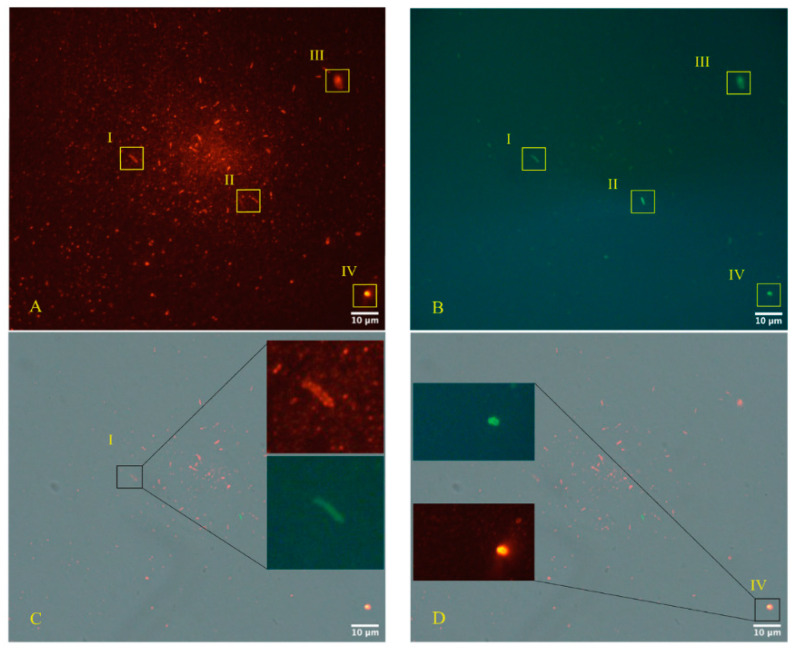
As reported by the authors in [34,35,36], (**A**–**D**) (Zeiss Axioplan 2, Axiocam 305 color, magnification 100×) show immunofluorescence staining of SARS-CoV-2 nucleocapsid protein (red light) versus Gram-positive bacteria (green light). (**E**) is the negative control, and (**F**,**G**) show a group of Gram-positive bacteria using fluorescence, derived from the stool bacteria culture of an otherwise healthy 18-month-old child (never suffering from illness and with healthy parents, but with SARS-CoV-2 at the time of collection, with the written consent of the parents) negative to SARS-CoV-2 molecular testing. However, the other primary antibody to the nucleocapsid protein is also included and does not show a red signal. The Roman numerals I, II, III, and IV, yellow rectangle, and the yellow arrows, indicate four Gram-positive bacteria (green light) infected by SARS-CoV-2 (red light). This data was presented with the kind permission of the authors [39,40].

**Figure 5 microorganisms-12-00643-f005:**
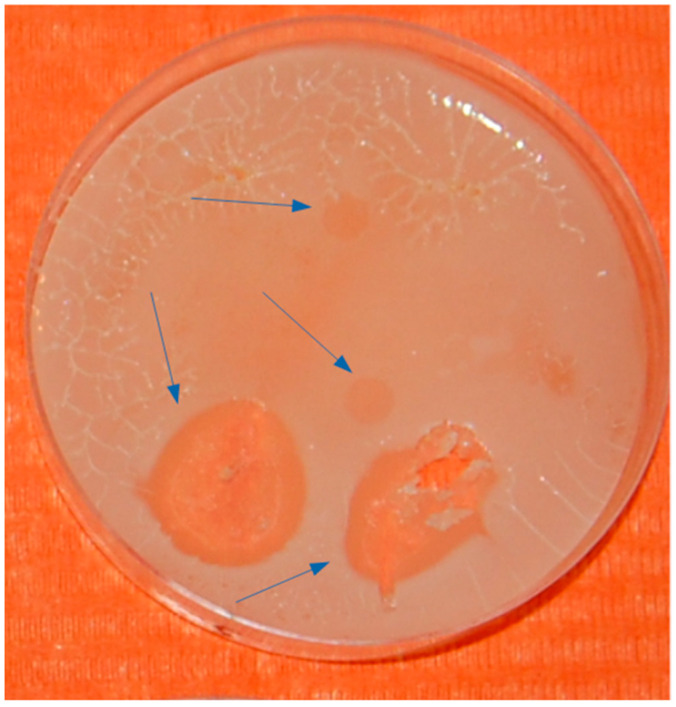
Lytic plaques (indicated by the blue arrows) on a plate where a bacterium of the genus Dorea contaminated different areas with different amounts of supernatant with SARS-CoV-2. The plaque area where sampling was performed was analyzed from a genetic point of view and the data were published in the paper by Petrillo et al. [38,39,40], where the emerging SARS-CoV-2 mutations in bacterial cultures are inferred.

**Figure 6 microorganisms-12-00643-f006:**
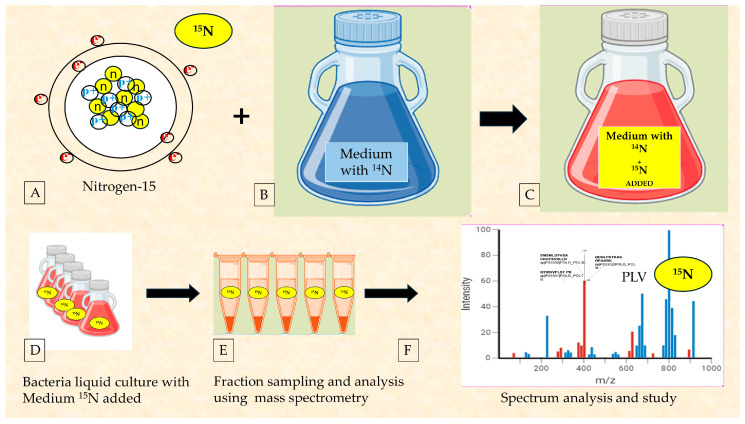
New data of Poliovirus replication in bacteria derived from the microbiome of elderly poliomyelitis patients with neurological disorders are presented. Bacterial cultures derived from the microbiome of Poliovirus patients were conducted by adding ^15^N nitrogen to the classical culture medium containing ^14^N nitrogen. After 7 days of culture, samples and their spectra were analyzed. (**A**): graphical representation of the nitrogen isotope, ^15^N, with one more neutron than the isotope ^14^N; (**B**,**C**): Culture broth for bacterial growth with ^14^N nitrogen medium after the addition of ^15^N isotope; (**D**,**E**): Schematic summary of the processing procedures of bacterial cultures fortified with ^15^N in which Poliovirus (PLV) is present; and (**F**): spectra of peptides of Poliovirus. Peptides containing labeled nitrogen are detected under neutral loss conditions and considering the proton rearrangement phenomenon.

**Figure 7 microorganisms-12-00643-f007:**
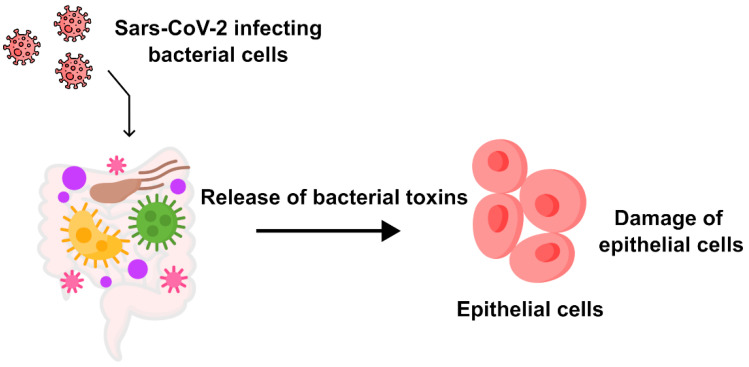
Pattern of toxin production by microbiome bacteria after being infected with SARS-CoV-2.

**Table 1 microorganisms-12-00643-t001:** Sabin reported data on poliovirus replication in feces despite the preview of Salk’s vaccination in the volunteers. Table V of [16]—*effect of feeding 10^6^ p.f.u. of type 1 attenuated poliovirus (l sc strain) to volunteers with (1) no homotypic antibody; (2) antibodies acquired from two doses of the Salk vaccine; and (3) naturally acquired antibodies.*

*Group*	*N° Volunteers*	*N° Excreted Virus*	*N° of Days Each Excreted Virus*	*Peak Virus Titers in Stool Log10, TCD50, per Gram*
*No antibody*	*11*	*11*	*10, 10, 10, 21+ *,*	*3.7, 3.7, 3.7, 3.7, 3.7, 3.7*
			*25+, 26+, 26+, 41, 77*	*4.2, 4.2, 4.2, 4.2, 4.2*
*Antibody after Salk vaccine*			*9, 10, 10, 13, 14,*	*2.7, 3.2, 3.4, 3.7*
	*8*	*8*	*21+, 26+, 42*	*4.2, 4.2, 4.7, 4.7*
*Naturally acquired antibody*				
	*8*	*1*	*10*	*4.2*

*(* 21+—Volunteer was fed another type of Poliovirus and the excretion of type 1 was interfered with).*

**Table 2 microorganisms-12-00643-t002:** The exclusion of the bacteriophages is obtained after extraction through the LC-SACI technique mass spectrometry [32,45,46,47] and their identification in UniProt bank data.

Putative Bacteriophage Sequences	Size and Form	Tail Yes/No
BK010646.1 TPA_exp: Bacteroides phage p00	50 to 70 nm in diameter and most tail lengths ranged from 120 to 200 nm	Yes
LR596903.1 Roseburia phage Shimadzu	Similar to a Syphoviridae	Yes
MT121960.1 Phage DP SC_2_H4_2017 DNA polymerase 1 and recombinase A genes, complete cds	Diameter. 75 nm, while the tail length is 125 nm	Yes
NC_024711.1 Uncultured crAssphage	Icosahedral capsids with short tails (type I, with head diameters of 76.5 nm and short tails; and type II, with a similar head size but head-tail collar structures and slightly longer tails)	Yes
NC_027980.1 Enterobacter phage phiKDA1,	Family T1 phage	Yes
NC_047910.1 Faecalibacterium phage FP_Epona	Phage with contractile tails	Yes
NC_047916.1 Faecalibacterium phage FP_oengus	Phage with long non- contractile tails	Yes

## Data Availability

Data are contained within the article.

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
