# Peer review of "Who Is the Intermediate Host of RNA Viruses? A Study Focusing on SARS-CoV-2 and Poliovirus"

_microorganisms, 2024, doi:10.3390/microorganisms12040643_

Round 1
Reviewer 1 Report
Comments and Suggestions for Authors
Major suggestions:
1. The manuscript jumps between several disjointed ideas without clear focus or logical flow. It would benefit greatly from reorganizing the content to have a more coherent narrative and focused argument. As is, the rationale and main point are unclear.
2. The significance and implications of the claims are not well explained. Even if the main hypothesis is proven that SARS-CoV-2 can infect bacteria, the authors need to better articulate why this matters and what the impacts are on our understanding of COVID-19 specifically.
3. The manuscript would be strengthened by removing tangential details and historical anecdotes that distract from the main thread. Staying focused on supporting the core hypothesis with data and critical analysis is advised.
Minor suggestions:
1. The writing style is very informal in parts and should be made more academic and precise throughout.
2. Some sections require further explanation for readers less familiar with the methodologies. For example, provide more context when introducing the nitrogen isotope test.
3. The figures and images require clearer captions and legends to interpret what is shown. Avoid asserting conclusions in the captions that the data may not fully support.
4. Carefully proofread for typos, grammar issues, and duplicate information. Streamline wordy or redundant passages.
5. Standardize the formatting and style to follow proper academic guidelines for a manuscript submission.
In summary, the manuscript requires significant revising to improve the clarity, evidence, organization, focus, and writing quality. Addressing the major issues would help strengthen the paper and make the authors' key claims more convincing and impactful.
Comments on the Quality of English LanguageThe writing style is very informal in parts and should be made more academic and precise throughout.
Author Response
R: We thank the reviewer for helping us and suggesting how to restructure the manuscript. We have undertaken the suggestions He/she has suggested and made the work more accurate. We have divided it into paragraphs and explained the radioisotope test well. We have also clarified the parallel between coronavirus and poliovirus better. We have also adjusted the references. We hope the reviewer can appreciate the changes as kindly suggested. We thank them for helping us.
R: English was revised by two native language authors of the paper: Dr. Valentina Viduto and Dr. Mark Fabrowski
Reviewer 2 Report
Comments and Suggestions for Authors
This commentary describes a very important point in SARS-CoV-2 infection, namely the evidence towards the potential replication of the virus in prokaryotic hosts, and that the spread of the disease could eventually be better controlled if this fact were considered. Within the manuscript, several points and aspects of this discovery are mentioned, such as an overview of the relevant results, a list of methods important in the identification, and even the comments on the reagents that have proven valuable in the research of this topic. I propose to correct 2 major points:
- The manuscript is at points difficult to understand because the language is unusual, sentences do not end or do not connect and there are inconsistencies present, such as in naming of the reagents or in the reference (order and contents).
- The manuscript should be reorganized, and the introductory paragraphs should list all the aspects that will be covered. Then the “overview of the results” and the discussion should follow this logic systematically, otherwise the important messages rendered are lost.
To that, I would recommend a careful re-read by all authors, as the level of the language used in the article appears to vary.
Please find below a list of further remarks which I hope you will find helpful.
Page 5 Figure 1: Text labels are too small.
Page 6: the reference has an unusual format?
Page 6: “with severe neurological conditions from poliomyelitis“ – please reword
Page 6: “How the authors in [34-36] described the figures 2,3,4 are one of the examples of many investigations obtained in the bacterial culture derived from sick patients to SARS-CoV-2 (Panels A-G)“ – I think you mean as the authors described in 34-36, the figures show the results of investigation of bacterial culture derived from patients with SARS-CoV-2, please correct the sentence for better clarity
,Figure 2: panel C is not described
Page 9: “It was also used as a primary antibody versus gram-positive Bacteria“: I think you mean gram-positive bacteria were stained with a primary antibody, please reword
Page 10, byline to Figure 4: For gentle concession of the authors – with kind permission of the authors is the more common expression
Table 2: “extrapolated after LC-SACI Technique at Mass spectrometry compared with UniProt bank data.“ – why extrapolated? Would this be extraction/identification?
Similar to a Siphoviridae should be similar to Syphoviridae
Page 11: “the culture of bacteria, added with“ should be the culture of bacteria with added N15, or similar
Page 12: “under the Culture“ – is this in the culture conditions?
Byline to Figure 6: “POL A”, abbreviations should be explained
Page 12: reference [78] is between 55 and 59, something with the order of inline citations is wrong
Page 13: Particularly visible from virus particle numbers 1 to 4 – please put the sentence into context.
Page 14:” antibodies act and function primarily at the epithelial interface“ – that is not necessarily true for all antibodies, what about the therapeutic antibodies? This statement is too general.
Page 14: BDI380 is a mouse and not antimouse antibody
Page 15: “increase in brightness given to fluorescence“ please reword, this expression is not clear
Page 15 citation: Luria andDulbecco, 194?
Figure 7: is this the final version of the Figure? This is toxin production, not toxicological production
Page 16: “The lytic plaque“ – sentences repeat
2.2 Considerations on the importance… break for the new paragraph missing
Page 16: “attenuated vision“ – attenuated version, do you mean?
Page 17: “finding an inert strain and basing mass vaccination“ – an inert strain that can serve as a base for mass vaccination?
Reference 77 is not mentioned in the text, and I could not find 40-45 in the text.
Comments on the Quality of English Language I would recommend a careful re-read by all authors, as the level of the language used in the article appears to vary. Several expressions are unusual and word order should be improved.
Author Response
Comments and Suggestions for Authors
This commentary describes a very important point in SARS-CoV-2 infection, namely the evidence towards the potential replication of the virus in prokaryotic hosts, and that the spread of the disease could eventually be better controlled if this fact were considered. Within the manuscript, several points and aspects of this discovery are mentioned, such as an overview of the relevant results, a list of methods important in the identification, and even the comments on the reagents that have proven valuable in the research of this topic. I propose to correct 2 major points:
- The manuscript is at points difficult to understand because the language is unusual, sentences do not end or do not connect and there are inconsistencies present, such as in naming of the reagents or in the reference (order and contents).
- The manuscript should be reorganized, and the introductory paragraphs should list all the aspects that will be covered. Then the “overview of the results” and the discussion should follow this logic systematically, otherwise the important messages rendered are lost.
To that, I would recommend a careful re-read by all authors, as the level of the language used in the article appears to vary.
Please find below a list of further remarks which I hope you will find helpful.
Page 5 Figure 1: Text labels are too small.
- we have addressed this issue. Now the format is larger. Thank you.
Page 6: the reference has an unusual format?
- We have corrected. Thank you.
Page 6: “with severe neurological conditions from poliomyelitis“ – please reword
- we have removed the phrase from this context and reintroduced it in the description as a paragraph of Figure 6.
"New data of Poliovirus replication in bacteria derived from the microbiome of elderly poliomyelitis patients with neurological disorders is presented."
Page 6: “How the authors in [34-36] described the figures 2,3,4 are one of the examples of many investigations obtained in the bacterial culture derived from sick patients to SARS-CoV-2 (Panels A-G)“ – I think you mean as the authors described in 34-36, the figures show the results of investigation of bacterial culture derived from patients with SARS-CoV-2, please correct the sentence for better clarity
- We have replaced it with: “The figures 2,3,4 are described by How the authors in [34-36]. They showed that Figure 2,3,4 are one of the many investigations obtained in the bacterial culture derived from sick patients to SARS-CoV-2 (Panels A-G), The panel H shows the presence of Nitrogen isotope 15N in the spike protein of SARS-CoV-2 after 7 days of a bacterial culture where the isotope was added to the nutrient broth.”
,Figure 2: panel C is not described
R: Post-embedding immunogold: rabbit monoclonal to SARS-CoV-2 Nucleocapsid protein antibodies ligating to the secondary anti-rabbit antibody 10nm gold-conjugated indicated the virus inside bacteria of the gut microbiome (Tecnai G2 Spirit BioTwin; FEI equipped with a VELETTA CCD digital camera (Soft Imaging Systems GmbH)).
Page 9: “It was also used as a primary antibody versus gram-positive Bacteria“: I think you mean gram-positive bacteria were stained with a primary antibody, please reword
R yes, thank you. We have corrected it.
Page 10, byline to Figure 4: For gentle concession of the authors – with kind permission of the authors is the more common expression
R: We have corrected.
Table 2: “extrapolated after LC-SACI Technique at Mass spectrometry compared with UniProt bank data.“ – why extrapolated? Would this be extraction/identification?
R: We have replaced it with “The exclusion of the bacteriophages is obtained after extraction through the LC-SACI Technique Mass spectrometry and identification in UniProt bank data”
Similar to a Siphoviridae should be similar to Syphoviridae
R: We have corrected in Syphoviridae.
Page 11: “the culture of bacteria, added with“ should be the culture of bacteria with added N15, or similar
R: done
Page 12: “under the Culture“ – is this in the culture conditions?
R: “and in the culture conditions it is replicated by the bacteria of the gut microbiome.”
Byline to Figure 6: “POL A”, abbreviations should be explained.
R: Pol A is a typo. We have replaced the whole of Figure 6 to make it more didactic and readable.
Page 12: reference [78] is between 55 and 59, something with the order of inline citations is wrong
R: We have corrected all references. Thank you.
Page 13: Particularly visible from virus particle numbers 1 to 4 – please put the sentence into context.
R We have replaced with: “Furthermore, in Figure 1A, you can see within the bacterium the viral particles, 1 to 4, at different densities from left to right.”
Page 14:” antibodies act and function primarily at the epithelial interface“ – that is not necessarily true for all antibodies, what about the therapeutic antibodies? This statement is too general.
R: The reviewer is absolutely right, we have made the sentence more specific: "Antibodies produced against viral pathogens, which have entry point on the mucosal surface, tend to localize to the epithelium as well."
Page 14: BDI380 is a mouse and not antimouse antibody
R: Yes, sorry, we have corrected it.
Page 15: “increase in brightness given to fluorescence“ please reword, this expression is not clear
R: Replaced with: “Parallel to the genetic aspect proposed by Petrillo et al. in another study [30-35], we demonstrated how also by fluorescence microscopy, by analyzing slides of bacterial culture samples, at predetermined intervals and up to 30 days, the bacteriophagic aspect of SARS-CoV-2 can be highlighted”
Page 15 citation: Luria andDulbecco, 194?
R: Luria and Dulbecco, 1949.
Figure 7: is this the final version of the Figure? This is toxin production, not toxicological production
R: Yes, sorry, It is toxins production. We have corrected it.
Page 16: “The lytic plaque“ – sentences repeat
R: We have removed one sentence. Thank you.
2.2 Considerations on the importance… break for the new paragraph missing
R: We edited the point better
Page 16: “attenuated vision“ – attenuated version, do you mean?
R: Attenuated virus. We have corrected
Page 17: “finding an inert strain and basing mass vaccination“ – an inert strain that can serve as a base for mass vaccination?
R: Yes, we have corrected. thanks
Reference 77 is not mentioned in the text, and I could not find 40-45 in the text.
R: We have corrected all references. Thank you
We thank the reviewer for helping us make the manuscript better and guiding us point to point. We hope that the revised version may be as he/she suggested.
Comments on the Quality of English Language
I would recommend a careful re-read by all authors, as the level of the language used in the article appears to vary. Several expressions are unusual and word order should be improved.
R: English was revised by two native language authors of the paper: Dr. Valentina Viduto and Dr. Mark Fabrowski.
Round 2
Reviewer 1 Report
Comments and Suggestions for Authors
The authors have well addressed my concerns.
Author Response
We thank the reviewer for improving our manuscript.
Reviewer 2 Report
Comments and Suggestions for Authors
The authors have considered several remarks and improved the manuscript. Unfortunately, there are still repetitions of sentences, and certain features of the Figures should be improved. Again, I suggest the re-read of the article by all co-authors, as the newly introduced text passages could be improved. Other remarks:
Line 215: “intestinal, nasal alveolar, and ocular“ – should probably be nasal, alveolar,…
Line 227-228: “that before the epithelial cells there is a thick layer of bacteria“. – in front of the epitheial cells…
Line 270, Figure 1: the size of the font in the Figure is too small and has not been improved
Line 347: Floridia, not Florida
Line 424: “derived from patients sick patients with SARS-CoV-2 (Panels A-G).“- please correct
Line 460: VELETA, not Veletta
Line 490, byline to Figure 3: „Panels E-G, pre-embedding immunogold“ – there are only panels E-F
Line 515: “was performed on the culture negative at molecular test to SARS-CoV2” – was performed with a culture with a negative molecular test for SARS-CoV2.
Line 519: “control of immunogenicity – probably you mean the control of specific reactivity
Line 528: “[BDI380] Genetex is an antimouse monoclonal“ – this is mouse and not antimouse antibody
Line 665: Isotopes are labelled e.g. 32 (32 in superscript) P – please correct throughout the manuscript
Line 666: Nitrogen in the molecular form N2 (2 in subscript)
Line 711: labels in the Figure should be 15N and not 15N2 (and so on), isotope is mostly added as 15N-aminoacids and has not much to do with N2. In the Panel A, there is quite clearly N and not N2.
Line 899: “The Rockland primary anti-body "Sars Nucleocapsid Protein Antibody [Rabbit Polyclonal]—500μg 200-401-A50 Rockland“ – this sentence is repeated (same as in the lines 541-542)
Line 944-945: the description of this antibody is repeated (see previous remark)
Lines 964-965: “that the bacteriophagic characteristic of SARS-CoV-2 can be highlighted. . The manuscript in [39] highlights“ – which bacteriophage characteristics and the word highlight repeats
Comments on the Quality of English LanguagePlease re-read the latest version - although the manuscript has improved in the level of English from the first version, there are weaknesses in the newly introduced text passages.
Author Response
We thank the reviewers for their diligence and support in this paper.
We note however some reviewer feedback from reviewer 2 does not correlate with in terms of line location. Probably there was a mistake in the upload of the manuscript.
However, we have resolved all his/her concerns and we thank him/her for the advice.
Reviewer 2 comment about line 215 is in line 188 of our last submission. The suggestion has been actioned. Thank you
Reviewer 2 comments about lines 227-228. This features in lines 203-204 of our last submission. The reviewer's suggestion has been actioned. Thank you.
Figure 1 font size in label adjusted.
Line 347 Reviewer 2 comment about Florida occurs in line 283 of our last submission. This has been corrected to Floridia -thank you.
Line 424 comment from Reviewer 2 occurs in line 349 of our last submission to you. The has been corrected - thank you.
Line 460 comment from Reviewer 2 occurs in line 363 of our last submission to you. We have corrected this spelling of VELETA. Thank you.
Line 490 in Reviewer 2 feedback occurs in line 393 of our last submission. We have amended the panel lettering byline accordingly. Thank you.
Reviewer 2 feedback on line 515 occurs in lines 416-7 of our last submission to you. We have taken your suggestion on board. Thank you.
Reviewer feedback on Line 519 occurs in our last submission to you at lines 415, 419, 432. We have taken the reviewer’s suggestion in the board and amended our wording. Thank you.
Reviewer comment on line 528 occurs in our last submission to you in line 422. This suggestion has been used to update our manuscript. Thank you.
Line 665: Isotopes are labelled e.g. 32 (32 in superscript) P – please correct throughout the manuscript. Line 665 isotope superscript number suggestion taken in board throughout the paper. Thank you. Line 665 reviewer comment occurs in our last submission in line 542. Corrected to subscript. Thank you.
Line 711: labels in the Figure should be 15N and not 15N2 (and so on), isotope is mostly added as 15N-aminoacids and has not much to do with N2. In the Panel A, there is quite clearly N and not N2. The image was corrected. Thank you.
Reviewer 2 comment on line 899 - In our last submission, the repetition was not at line 541, but at 432 and 715-721. This repetition has been deleted and replaced by another sentence. Thank you.
Reviewer 2 suggestion of lines 944-945 being repeated - we do not see this in our last submission but at line 748. It was already corrected. Thank you.
Reviewer 2 comment re lines 964-965 exists in lines 762-763 of our version. This does match our last submitted version but we have changed the second highlight to demonstrate. Thank you.
We thank the reviewer for all the attention he paid to our manuscript.